# Country Economic Security Monitoring Rapid Indicators System

Sergei N. Mityakov [1], Evgenii S. Mityakov [2,*], Andrey I. Ladynin [2] and Ekaterina A. Nazarova [1]

[1] Institute of Economics and Management, Nizhny Novgorod State Technical University n.a. R.E. Alekseev, 603155 Nizhny Novgorod, Russia; snmit@mail.ru (S.N.M.); k-romanova@list.ru (E.A.N.)
[2] Institute of Cybersecurity and Digital Technologies, MIREA—Russian Technological University, 119454 Moscow, Russia; ladynin@mirea.ru
* Correspondence: mityakov@mirea.ru

**Abstract:** Time series analysis is a method of key importance for systems of various hierarchies' economic security studies. This article's main goal is to develop an economic security rapid indicators system, introducing threshold values and utilizing indices with a one-month sampling period, and its approbation during Russia's economic security operational monitoring. In order to develop such a system, the authors accumulated economic security world experience including reliability, visibility and tree structure principles. The authors' monitoring system includes four spheres: real economy, social, monetary and foreign economic, each of which contains three indicators. In order to organize economic security monitoring, it is proposed to use the index method, which converts indicators into a dimensionless form with integral values in subsequent calculations. Based on integral indices values, the economic security generalized index is synthesized, which can be used to analyze a system's development trends. We present economic security normalized indicators and integral indices dynamics for the years 2020–2022, which show two crises dynamics. The first is due the COVID-19 pandemic, while the second is associated with economic sanctions against Russia, implemented in 2022. The proposed economic security operational monitoring indicators system can be used effectively in the government's practical tasks in order to ensure the required level of economic security. This is especially true for rapid diagnosis of crisis phenomena in countries and individual regions.

**Keywords:** economic security; fast economic security indicators; threshold value; economic crisis; integral indices; operational monitoring; time series analysis





## 1. Introduction

This article presents an economic security rapid indicators system with a single month reference period and threshold values for analysis of emerging crisis situations and prediction for prompt decision-making at various economy hierarchical levels. The objects of the study are the crises situations in the economy, including collapses in the commodity and financial markets, epidemics and pandemics, natural disasters and catastrophes with critical economic consequences, and social, criminal and terrorist tensions and dangers.

Time series dynamics analysis and evolution forecasting are of great importance for managing processes in social and economic systems (Andrianova et al. 2020). Systems' engineering methods nowadays make it possible to create real object information models, supplemented by virtual components, and, vice versa, i.e., virtual object models, supplemented by real components (Rechkalov et al. 2023). Scientific works appear in economic research devoted to analysis and forecasting problems using machine learning and artificial intelligence tools (Mityakov and Mityakov 2020; Coulombe et al. 2022). In this work, despite rapid indicator usage, we are dealing with fairly short time series, which makes it difficult to use both classical econometric methods for analysis and forecasting, and relatively modern methods based on machine learning.

The work utilizes the authors' earlier developments, including economic security operational monitoring with rapid indicators, a two-threshold indicators system and the index method. The main scientific elements are the detailed analysis methodology of the last two economic crises, and the economic security generalized index summary analysis for the 1997–2022 period. This makes possible not only to identify the characteristic features of each crisis, but also to formulate a number of hypotheses related to predicting possibility. These hypotheses include RTS index dynamics usage as a crisis harbinger, as well as changes in the statistical properties of indicators during these periods.

The rest of the paper is organized as follows. Section 2 contains the literature review, Section 3 outlines the methodology's key components: approaches, stages and analysis methods. Section 4 presents the main results obtained during the analysis of Russian economic security rapid indicators dynamics for the regarding period. The results obtained in the authors' earlier studies are also discussed here. In particular, the economic security generalized index dynamics for the period 1996–2022 are given, and some considerations are given related to crises' prediction possibility. Section 5 contains practical conclusions, limitations and further perspectives.

## 2. Literature Review

### 2.1. Economic Security Concept

The beginning of research into economic security issues is associated with the Great Depression in the United States and the New Deal by F.D. Roosevelt. In 1934, the US President created the Federal Committee on Economic Security, which dealt mainly with unemployment and the protection of citizens' economic interests. During the Cold War, economic security was viewed through the superpowers' economic support prism, the arms race, and was interpreted from the standpoint of special services' participation. It is advisable to consider foreign countries' economic security concepts from a standalone viewpoint, since each state has its own characteristic conditions and features.

In Russia, the main national and economic security issue was due to the severe systemic crisis in the 1990s, which ended in a 1998 default. In 1994, Academician L. Abalkin, in the periodical *Voprosy Ekonomiki* [*Economics Tasks*], initiated the state's national security, ensuring discussions on scientific and practical substantiating issues (Abalkin 1994). The monograph by V. Senchagov discusses the economic security essence concept, defined as "such economy and power institutions condition, which provides guaranteed national interests protection, countries socially oriented as a whole development, sufficient defense potential even under the most unfavorable conditions for internal and external processes development" (Senchagov 2010).

In the United States of America, the economy plays a role in providing the resources needed to keep citizens safe, reduces unemployment and keep households economically secure (Nanto 2011). The American researcher B. Buzan understands the term "economic security" as the state of the economy, which ensures the economic well-being of subjects participating in public relations (Buzan 1991).

France's national security assessment is based on the continental three-level system "individual–society–state". Personal security includes two criteria: French citizens' rights and freedom protection levels, and society economic security. Society security covers a degree of population patriotism and stable citizens' future confidence. State security is the ability to protect French borders from encroachments from outside and economic system independence (Samogin and Galanova 2021).

P. Hough identifies three main approaches in economic security foreign studies. In his opinion, economic security can be achieved: from a liberal point of view, through more intensive globalization; from the mercantilists' viewpoint, through less globalization, while Marxists assume global radical changes (Hough 2014). M. Kahler believes that globalization has "undermined" the traditional definition of economic security, which focused on economic vulnerability to other states. At the same time, globalization has caused its redefinition in the associated risks of non-state actors' cross-border networks,

as well as new environment economic volatility (Kahler 2004). A. Posen and D. K. Tarullo believe that globalization is a process that shapes the international environment and undermines the old definition of economic security, forcing it to be redefined (Posen and Tarullo 2017). C. Lessmann studied the influence of inter-regional inequality within countries on internal conflicts. In his opinion, regional inequality increases internal conflict risks, and therefore creates a threat to economic security in the regions (Lessmann 2013). A. Ignatov describes European state security as the ability to implement policies and strategies effectively to achieve desired goals in the face of external or internal threats. As for indicators determining economic security levels, it is proposed to use the total-debt-to-GDP ratio, real GDP growth rate, fixed capital accumulation, productivity per resource unit used, high technologies and public administration effectiveness (Ignatov 2019).

S. Muratova considered methodological approaches to ensuring economic security and European Union countries' internal market protection. Measures for developing and implementing European countries' economic security have been systematized, and their application in the Republic of Uzbekistan have been analyzed (Muratova 2020).

The COVID-19 pandemic became a significant impetus for the popularity of economic security ideas, as it led to a large-scale economic crisis, which consequently strongly affected households in Europe and America (McCormick et al. 2020). Another factor affecting the economic security of Europe is the geopolitical situation. Examples include trade wars between states and transnational companies' market capture (Meijnders and Merel 2019).

The aggravated geopolitical situation dictated the emergence of new studies on economic security. For example, in order to protect the European Union's security and interests, J. Borrel indicates the risks that need to be closely monitored (Borrell 2023). These include supply chains' sustainability risks, critical infrastructure's physical and cyber security, risks associated with technological safety and technology leakage, and turning economic dependencies into coercion weapons risks.

*2.2. Economic Security Monitoring*

World experience analysis shows that effective monitoring organization and conduct are key elements in ensuring economic security. In developed countries, regular monitoring has already become a real management tool since the 1990s. Contemporary scientific studies are devoted to economic security monitoring. For example, K. Borio and F. Lowe, based on 34 industrialized countries and emerging markets countries with an average income level, examined the effect of the boom in prices for assets' impact, credit or investment on the financial crisis development. They empirically determined indicators' threshold values, the achievement of which allowed them to talk about the crisis beginning (Borio and Lowe 2022). J. Grikietytė-Čebatavičien conducted a financial security assessment of EU countries. The financial security composite index was used, which is based on the following sub-indices: human development, financial globalization development, economic freedom and financial stress at the country level. The results showed that developed EU countries have high financial security (Grikietytė-Čebatavičienė 2021).

O. Hrybinenko, O. Bulatova and O. Zakharova analyzed countries' solvency levels based on financial indicators' multidimensional methodological assessment tools, the result of which is integral indices construction corresponding to economic security level. The proposed approach allowed the determination of solvency levels as follows: critical, dangerous, unsatisfactory, safe and optimal (Hrybinenko et al. 2020). Analysts at Women's Policy Research developed the Basic Economic Security Tables (BEST) Index. This study found that incomes above the federal poverty line were insufficient to provide basic economic security (Suh et al. 2018). The paper by Yuan Guanghui, Xie Fei and Tan Huiling focuses on building an early diagnostic system for economic security using cloud computing and data mining technologies. The authors' model can adaptively assess the economic security state (Yuan et al. 2022).

In Russian, economic security issues and entities ensuring, monitoring and managing conceptual apparatus have been worked out in more detail. In 2011, an updated indicators

list for economic security analysis was published (Senchagov 2011). In order to monitor the country's economic security, four spheres were identified: social, monetary and financial, real economy and foreign economic spheres. In 2017, the economic security strategy was adopted, which included 40 indicators (Decree of the President of the Russian Federation No. 208 2017).

A significant number of scientific works are devoted to regional systems' monitoring of economic security. We can highlight papers by Russian Academy of Sciences Ural Branch Economic Security Center researchers (Kuklin et al. 2013; Tatarkin et al. 1996). We also note Nizhny Novgorod State Technical University. n.a. R.E, Alekseev scientists' papers (Mityakov et al. 2013, 2020). T. Rudakova, I. Sannikova and O. Rudakova substantiated the basic elements of regional economic security. In order to achieve this, structural analysis and mathematical modeling methods were used. As a complex indicator that determines a region's economic security, the gross regional product per capita was proposed (Rudakova et al. 2018).

E. Mityakov showed that regional economic security monitoring requires an integrated approach, whose key factor is a single methodology based on current methods, models, approaches and tools. An economic security regional system's main elements were identified, and a monitoring algorithm was proposed (Mityakov 2018). I. Averina and M. Buyanova presented an information and analytical system for regions' economic security monitoring, containing the following modules: centralized data storage, risk management, regional development security modeling and managerial decision support (Averina and Buyanova 2019).

M. Rudenko introduced an economic security system for Perm, one of the Russian Federation's industrial regions. Economic security main indicators of the Perm Territory were analyzed in comparison with other Volga Federal District regions (Rudenko 2019). E. Husainova, L. Urazbakhtina, N. Serkina, E. Dolonina and O. Filina formulated an algorithm for assessing economic security risks in a management system, achieving income from risks by levels and types of their impact based on the authors' developed scale (Husainova et al. 2019). V. Manyaeva, O. Naumova and S. Sotskova enhanced the economic security monitoring mechanism stages (Manyaeva et al. 2019). B. Kozicki, M. Górnikiewicz and M. Walkowiak explored the impact of the COVID-19 pandemic on Russian and European countries' economic security levels. The multifaceted analysis of oil prices was carried out before and after the pandemic announcement. Then authors analyzed the demand for oil and air travel. For analysis, categorized linear histograms were used, with indices from the US Federal Statistical System (Kozicki et al. 2020). T. Satsuk and O. Koneva proposed their own approach towards health industry economic security assessing and monitoring. The need to ensure economic security was heightened during the COVID-19 pandemic, when the health industry took the greatest toll. A methodology has been developed that uses a complex state's indicator and its assessment scale (Satsuk and Koneva 2022). N. Kuznetsov has developed a system for monitoring economic security, which is a means of countering challenges and threats in the socio-economic sphere. The system has a three-level architecture, including data contours, risk and risk analysis, and control action generation (Kuznetsov 2019).

V. Starovoitov and N. Starovoitov summarized conceptual approaches to the consideration of the federal risk management system in Russia. Such an indicators system allows the usage of systematic analysis approach tools, dynamics simulation of complex socio-technical and socio-economic objects in high uncertainty conditions, as well as decision-making on the accuracy of government responses in case of risks and economic security consequences (Starovoitov and Starovoitov 2019). M. Barsukova, A. Nikolaeva, T. Stolyarova and L. Fedorova substantiated the pattern for economic security new technologies' monitoring, introducing financial transactions' complications, innovations and digital statistics development. Special opportunities for transforming economic security monitoring, based on digital protocols for express analysis of crisis situations and threat

prediction of economic security information arrays formation, are presented (Barsukova et al. 2019).

Separately, we should dwell on the Nizhny Novgorod economic security operational monitoring school. Therefore, for operational forecast formation and analysis accuracy, so-called "rapid" indicators are used, updated monthly or each quarter. The first works, including Russia's economic security dynamic analysis using rapid indicators, were carried out by V. Senchagov and S. Mityakov (Senchagov and Mityakov 2013). Later, Nizhny Novgorod authors published several more works on this topic (Mityakov and Mityakov 2021; Mityakov et al. 2019). This study is a continuation of this cycle.

## 3. Methodology

### 3.1. Economic Security System Structure

The economic security system structure consists of the following methodological sections.

1. Legal framework:

   - Russia's economic security strategy for the period up to 2030;
   - Russia's national security strategy.

2. Russia's national interests in the economic sphere:

   - Economy development and ensuring the country's economic security;
   - Creating personal development conditions and improvement of citizens' life quality;
   - Ensuring technological sovereignty;
   - Russia's entry into leading countries' ranks in terms of gross domestic product.

3. Economic security threats:

   - Internal threats: economy's state regulation inefficiency, lack of innovative development and interests balance violation in the economy developing most effective ways search;
   - External threats: high volatility in world energy prices, significant fluctuations in the national currency exchange rate, capital outflow over its inflow excesses, increase in corporate debt, raw materials export overload and the economy's significant dependence on imports.

4. Economic security operational monitoring (ESOM) concept definition:

   ESOM stands for the country's economic security rapid indicators constant monitoring process and Russian Federation constituent entities, including necessary data collection, economic situation dynamic processes analysis, and identification of development trends and threats forecasting.

5. The main tasks of the ESOM:

   - Reliable information collection, operational monitoring organization and search for reliable data on the state of the country's economic security in dynamics;
   - Country's economic security analysis and assessment and results comparison with certain criteria;
   - Filling and periodic updating ESOM databases in order to develop adequate mechanisms for neutralizing threats;
   - Identification of forecasting threats and timely crisis phenomena "harbingers" in the country's economy;
   - Recommendations for authorities' preparation at various hierarchy levels for the country's economic security system operational management purpose.

### 3.2. Step-by-Step ESOM Procedure

The following is a step-by-step monitoring procedure:

1. Monitoring problems statement, immanent tools and requirements definition.

2. Necessary up-to-date information search. This stage includes the indicators system and its threshold levels choice. In this work, the system proposes a short-term indicators set with a one-month sample discretization, which is published in official statistical collections.

3. Initial information on economic security indicators' transformation. In addition to the main system, additional ones are used that are necessary for evaluation and precursors search.

4. Index method implementation (Senchagov and Mityakov 2011), adapted by us for the ESOM problem. Converting indicators to a dimensionless form. After the transformation, indicators retain trends in their dynamics and vary within the same scale, which opens up the possibility for their analysis using radar charts

5. Indicators' interaction patterns analysis, both integral and generalized economic security indices.

### 3.3. Economic Security Rapid Indicators System

Table 1 shows Russia's economic security rapid indicators system, which we used when organizing the ESOM.

**Table 1.** Russia's economic security rapid indicators.

| No. | Indicator Name | Critical Value | Target Value | Reference |
|---|---|---|---|---|
| Real economy sphere | | | | |
| 1 | GDP physical volume index | 101.5% | 104% | http://www.gks.ru accessed on 17 April 2023 |
| 2 | Industrial production index | 100% | 106.5% | http://www.gks.ru accessed on 17 April 2023 |
| 3 | Fixed investment index | 100% | 105% | http://www.gks.ru accessed on 17 April 2023 |
| Social sphere | | | | |
| 4 | Labor market tension coefficient | 3 ppl. | 1 ppl. | http://www.gks.ru accessed on 17 April 2023 |
| 5 | Real disposable money income index | 100% | 104% | http://www.gks.ru accessed on 17 April 2023 |
| 6 | Retail turnover index | 100% | 103.5% | http://www.gks.ru accessed on 17 April 2023 |
| Monetary and financial sphere | | | | |
| 7 | Goods and services monthly imports' coverage by gold and foreign exchange reserves | 3 month | 22 month | http://www.cbr.ru accessed on 19 April 2023 |
| 8 | Consumer price index, % | 113% | 104% | http://www.gks.ru accessed on 17 April 2023 |
| 9 | Net capital outflow, % compared with goods and services exports | 25% | 9% | http://www.cbr.ru accessed on 19 April 2023 |
| Foreign economic sphere | | | | |
| 10 | External debt, % of GDP | 50% | 30% | http://www.cbr.ru accessed on 19 April 2023 |
| 11 | Export volume index | 102% | 106% | http://www.gks.ru accessed on 17 April 2023 |
| 12 | Import volume index | 102% | 106% | http://www.gks.ru accessed on 17 April 2023 |

The system contains the following socio-economic system functioning spheres of the country: real economy sphere, social sphere, monetary and financial sphere and foreign economic sphere. The choice is consistent with economic security monitoring classical methodology adopted by the Russian Academy of Sciences Economics Institute (Senchagov 2011).

Each sphere includes 3 indicators, characterizing its economic security from different positions. Mostly, these indicators are included in the existing list, defined in the Russian

Federation economic security strategy. For each of them, critical and target values are determined. These values, as well as sources of information, are given in the table. The choice of indicators is based on the presence of official information sources, the possibility of using them for operational analysis, interdependence absence and economic security, ensuring relevance. Let us consider in more detail this indicators system and its impact on economic security levels.

1.  The GDP physical volume index is calculated as the ratio of the current GDP volume to the GDP volume in the corresponding period of the previous year, multiplied by 100%. It is the country's economic growth indicator.
2.  The industrial production index is an indicator for production volume analysis and is considered as the ratio of the current to the previous year periods' production volume, multiplied by 100%. It is an indicator of the country's industrial production level growth (fall).
3.  The fixed investment index is a relative indicator that characterizes the change in the volume of capital investments in the current period compared with the corresponding period of the previous year, multiplied by 100%. It represents the main factors of economic growth and technological development.
4.  The labor market tension coefficient indicates the number of unemployed people per one declared vacancy. In fact, it determines the total number ratio of the unemployed population to the number of vacancies, and is the country's personnel security indicator.
5.  The index of real disposable money income is a relative indicator that characterizes the change in real disposable money income in the current period compared with the corresponding period of the previous year, multiplied by 100%. It is an important indicator that characterizes the country's social sphere state.
6.  The retail turnover index is a relative indicator that characterizes the change in retail trade volume in the current period compared with the corresponding period of the previous year, multiplied by 100%. It is population consumer demand changes indicator, which is an important economic security indicator.
7.  Goods and services monthly imports coverage by gold and foreign exchange reserves shows how many months it takes to cover imports using gold and foreign exchange reserves. In other words, how long a country can pay for imports solely from reserves. It is the financial airbag of the state. It is accumulated in order to overcome crisis moments more easily, to stimulate the economy in difficult periods.
8.  The consumer price index is a classic inflation indicator. It shows the change in prices for consumer goods and services, fixed at a constant quantity, and properties purchased, used or paid for by the population in the current period compared with the corresponding period of the previous year, multiplied by 100%.
9.  The net capital outflow, a percentage compared with goods and services exports, is the difference between capital exports and imports in the country in the current period in relation to the export volume, multiplied by 100%. It shows what export earnings can be taken out of the country, maintaining an economic security level.
10. External debt is an external public debt sum, and indicates the country's ability to repay its debts. External public debt has long ceased to be Russia's economic security problem and external corporate debt sum.
11. The export volume index is calculated as the goods and services exports volume to the exports volume ratio, determined in the corresponding period of the previous year, multiplied by 100%. It shows the country's potential export results.
12. The import volume index is calculated as the goods and services imports volume to the imports volume ratio, determined in the corresponding period of the previous year, multiplied by 100%. It demonstrates the country's necessary provision level for the production of external resources.

*3.4. Economic Security Indicators' Critical and Target Values*

Critical values are those key indicators that distinguish between the economic security's normal state and a state characterized by an increased threats manifestation. To determine such values, expert assessments, the analogies method and mathematical methods are usually used. Since critical values are not the main goal for this publication, we used data from various sources (Senchagov 2011; Krivorotov et al. 2019; Medvedenko 2020). Indicator targets are desired values that need to be achieved to ensure the country's strategically significant economic security level. Target values were determined based on Russian Federation Economic Development Ministry recommendations (Interfax 2019).

*3.5. "Index Method" Implementation*

Let us consider the index method for the main stages of the ESOM application.

3.5.1. Indicator Conversion to a Dimensionless Form

In this case, the so-called traffic light model was used, which uses three risk zones. If the indicator value lies below the critical level, then it falls into the "red" zone; if it between the critical and target levels, then it falls into the "yellow" zone; and, if it is above the target level, it falls into the "green" zone. The economic security operative monitoring system contains both "direct" and "reverse" indicators. The "direct" indicators' growth has a positive impact on the country's economy (for example, the "GDP physical volume index"). Meanwhile, "reverse" indicators' growth, on the contrary, reduces the economic security level (for example, "annual inflation rate").

In order to realize the joint analysis possibility, indicators are reduced to a dimensionless form and uniform possible changes limits. It is proposed to use the arc tangent, Formula (1), which a normalizing function and can be applied equally to both "direct" and "reverse" indicators:

$$y = \frac{3}{\pi}\left(\frac{\pi}{2} + \tan^{-1}\left(\left(\tan\frac{\pi}{6}\frac{2}{b-a}\right)\left(x - \frac{a+b}{2}\right)\right)\right), \tag{1}$$

The following designations are used: $x$ is an initial indicator and has natural values, $y$ stands for a converted indicator, and $a$ and $b$ are critical and target natural values, respectively.

After normalization, all indicators are converted to "direct", while critical values are displayed at the level of $y = 1$, and target values at the $y = 2$ level, so the converted indicators' acceptable values range includes the segment [0; 3].

3.5.2. ESOM Indicators' System Integral Indices Using Spheres and Economic Security Generalized Index Synthesis

To analyze trends in the various spheres of economic security, integral indices were synthesized:

$$Z_i = \sum_{j=1}^{3} p_j y_{ij}; \ \sum_{j=1}^{3} p_j = 1, \tag{2}$$

where $y_{ij}$ is the $j$-th indicator of the $i$-th sphere, and $p_j$ is its weight coefficient, which reflects expert opinions, statistical observations or decision maker demands. In this particular study example, we set all weights equal and the sum is 1.

The economic security generalized index was calculated as the sum of all integral indices, taking into account their significance:

$$Z = \sum_{i=1}^{4} s_i Z_i; \ \sum_{i=1}^{4} s_i = 1, \tag{3}$$

where $s_i$ is the economic security $i$-th sphere weight, and $Z_i$ represents the integral index and is calculated by Formula (2).

Indices, as well as normalized indicators, have valid values range from 0 to 3. Special attention should be paid to indicators' weight coefficients and economic security spheres' finding issues. Various approaches can be used for this. Equal weights or coefficients for various indicators included in the index are a possible approach to building generalized indices. This approach is easy to implement and the results are relatively easy to interpret, but it has its drawbacks. First, equal weighting does not take into account the fact that some indicators may be more important to or more influential on the issue at hand than others. Second, generalized indices with equal weights may not take into account complex relationships between indicators or data features. In real practical problems, more complex weighting methods are often used, which take into account indicators' importance based on expert opinion, statistical methods, sensitivity analysis and other approaches. This allows the building of more accurate and adequate generalized indices that reflect the real situation and can be useful for decision-making. Another option is to assign more weight to the indicator that is furthest away from the threshold, since that indicator may be the most dangerous in the current situation. In our case, expert assessments did not reveal any tangible benefits to ensuring contribution from both individual indicators and economic security individual spheres. Therefore, in all calculations, corresponding indicators' weight coefficients and economic security spheres were assumed to be equal.

### 3.6. Additional Indicators

In addition to the main indicators (Table 1), exchange groups and additional indicators were used in the course of the analysis. They were considered by us as exogenous (external) model parameters. Oil price, the RTS index and the dollar-to-ruble ratio were chosen as such indicators. The indicators' monthly average values were used.

### 4. Results and Discussion

Figures 1–12 show Russia's ESOM normalized indicator dynamics for the 2020–2022 period. Two crises are clearly visible here. The first one is associated with the start of the COVID-19 pandemic, which coincided with the sharp drop in oil prices in early March 2020. The crisis was accompanied by a drop in most economic security indicator values. The exception was the gold-reserves-to-exports-volume ratio (Figure 7), which had almost no changes, remaining in a comfortable "green" zone. The consumer price index value (Figure 8) gradually decreased (which corresponded to a slow growth in inflation), and, for the 2020–2021 period, gradually moved from the "green" to the more problematic "yellow" zone. Also, the external-debt-to-GDP ratio remained virtually unchanged (Figure 10), remaining on the "green" and "yellow" zones' border. As for the net capital outflow, compared with goods and services exports (Figure 9), its dynamics in 2021 were sharply non-stationary, almost constantly being within the "yellow" zone. The economic recovery was already noted by May 2021 (exports and imports indices—by February 2021). At the same time, inflation continued to rise in 2021 and capital outflow from the country continued.

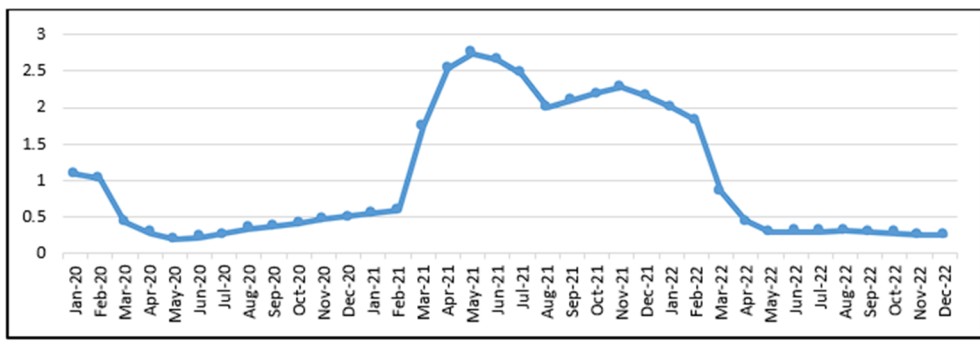

**Figure 1.** GDP physical volume index dynamics.

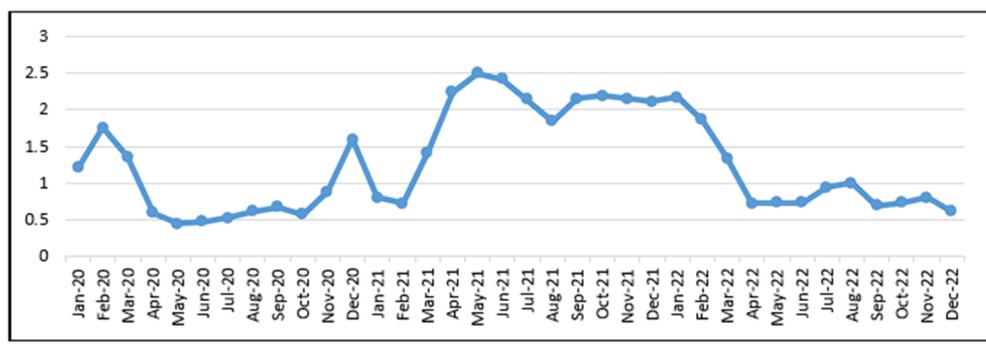

**Figure 2.** Industrial production index dynamics.

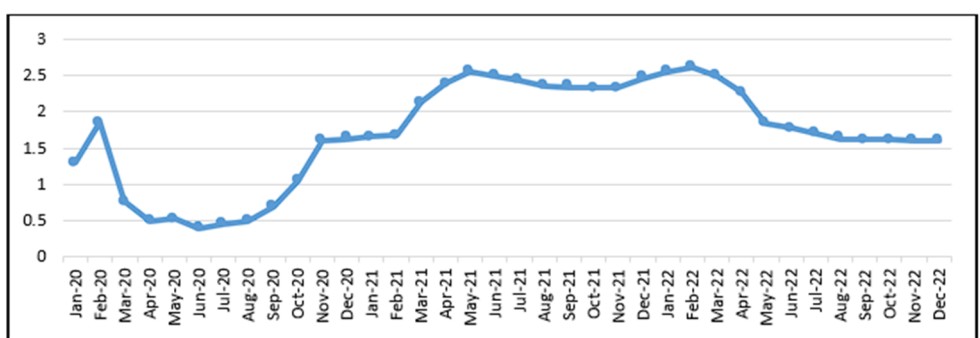

**Figure 3.** Fixed assets investment index dynamics.

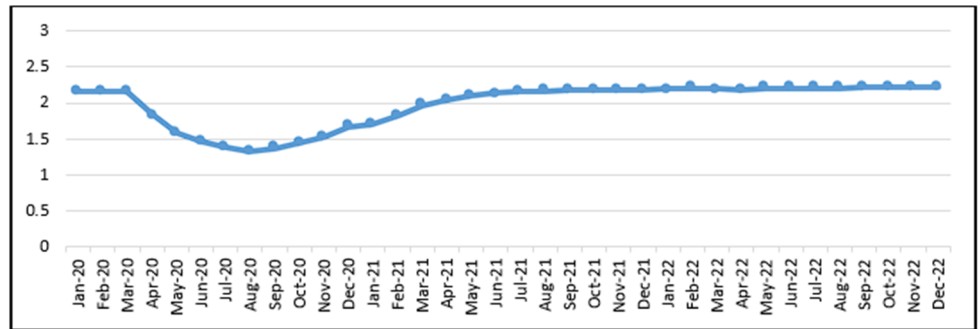

**Figure 4.** Labor market tension coefficient dynamics.

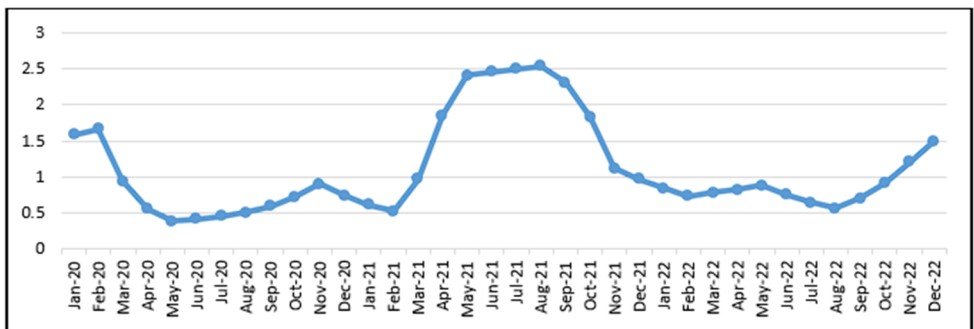

**Figure 5.** Real disposable money income index dynamics.

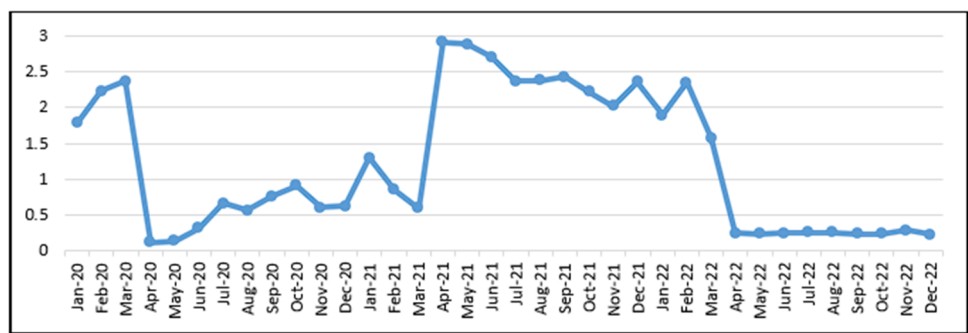

**Figure 6.** Retail turnover index dynamics.

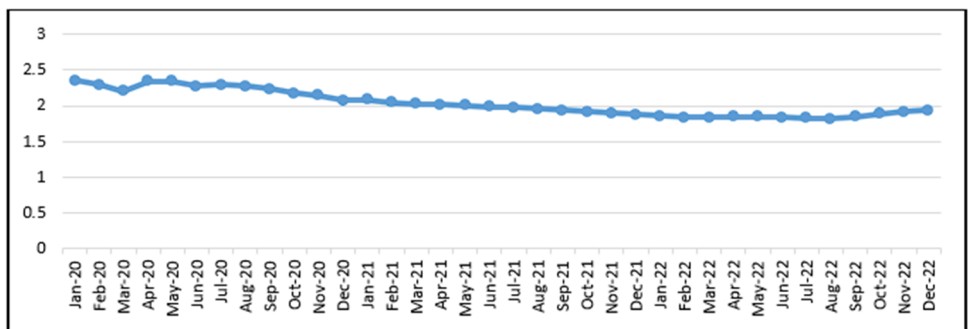

**Figure 7.** Gold-and-foreign-exchange-reserves-to-exports-volume ratio index dynamics.

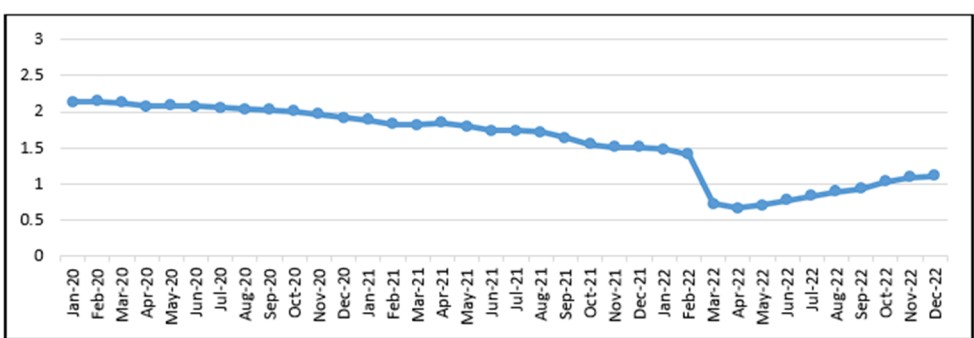

**Figure 8.** Consumer price index dynamics.

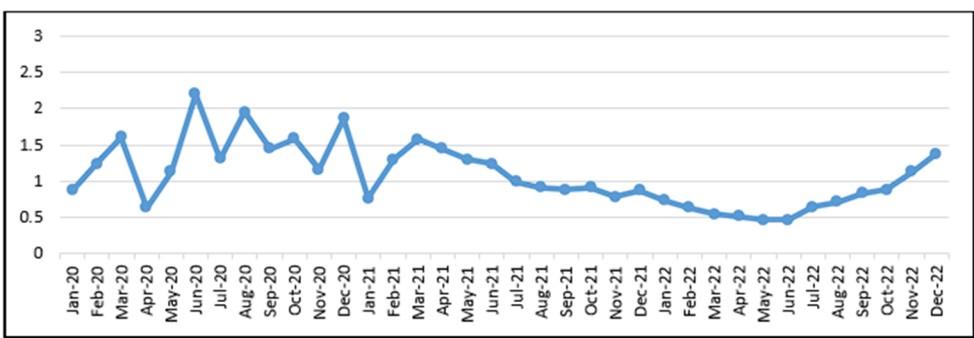

**Figure 9.** Net capital outflow, % compared with goods and services exports.

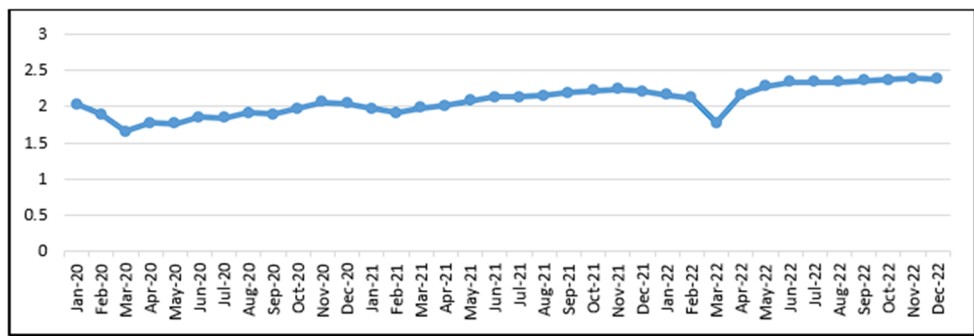

**Figure 10.** External debt, % of GDP dynamics.

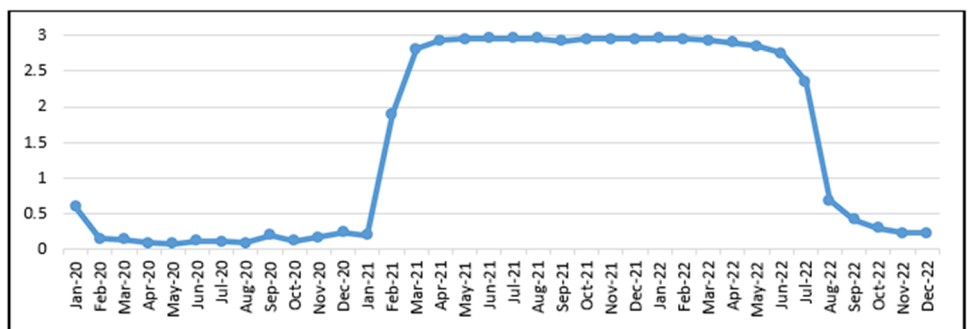

**Figure 11.** Export volume index dynamics.

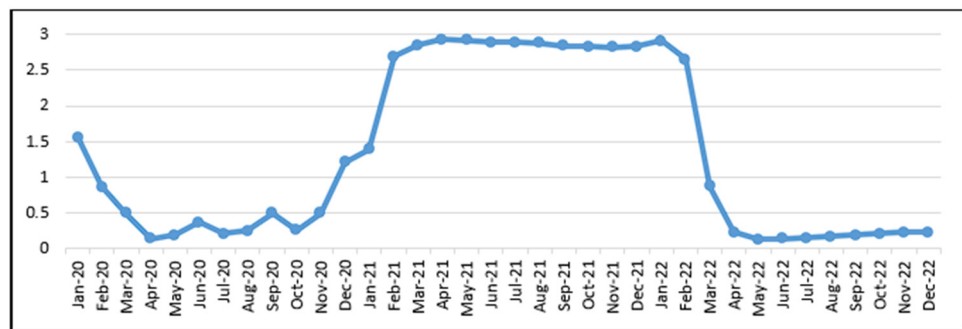

**Figure 12.** Import volume index dynamics.

The second crisis is associated with the introduction of economic sanctions against Russia in connection with the current foreign policy situation. Note that, even before February 2022, at the end of 2021, the real disposable income index began to fall again. After February 2022, there was a drop in such indicators as imports, GDP physical volume, the industrial production index, the inflation rate and retail trade turnover. A little later, there was a drop in fixed assets and exports investments. By the end of 2022, the following indicator values had significantly improved: inflation, capital outflow and real disposable income.

Figures 13–16 show Russia's ESOM spheres' integral indices dynamics in 2020–2022. The real economy sphere and social and external economic spheres present very similar dynamic pictures, having recession and recovery periods corresponding to the development of the crises mentioned above. The monetary and financial sphere dynamic picture (Figure 15) is significantly different. It includes two conditional sections. The first one, from January 2020 to February 2022, illustrates a weak fall in the integral index within the "yellow" zone. Starting from February 2022, the linear trend sign has changed to positive, which is associated with a slowdown in inflation, a decrease in external debt and an increase in gold and foreign exchange reserves. Figure 17 shows the generalized economic security

index dynamics for the period 2020–2022, and Figures 18–20 show additional indicator dynamics (RTS index, and ruble against dollar and oil prices, respectively) for the same period. It should be noted that, for the first time during the observation period, the 2022 crisis developed against the backdrop of rising oil prices.

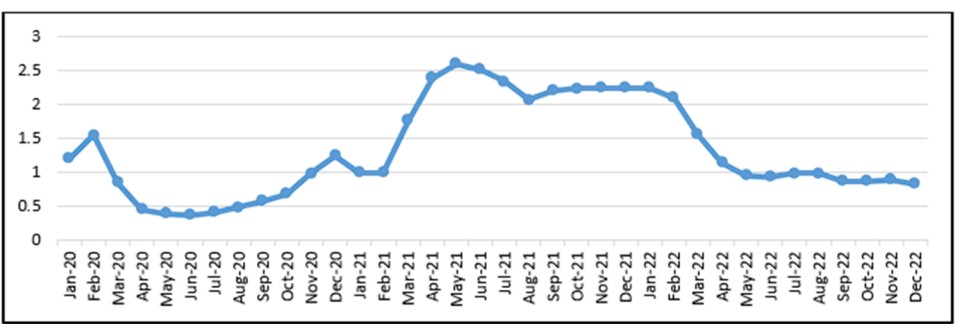

**Figure 13.** Real economy sphere integral index dynamics.

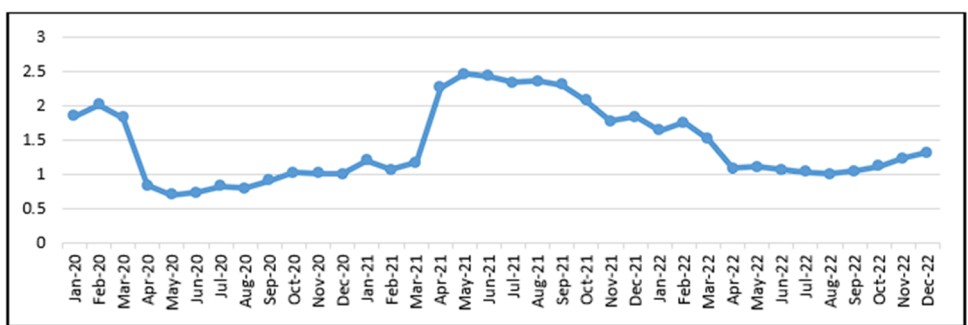

**Figure 14.** Social sphere integral index dynamics.

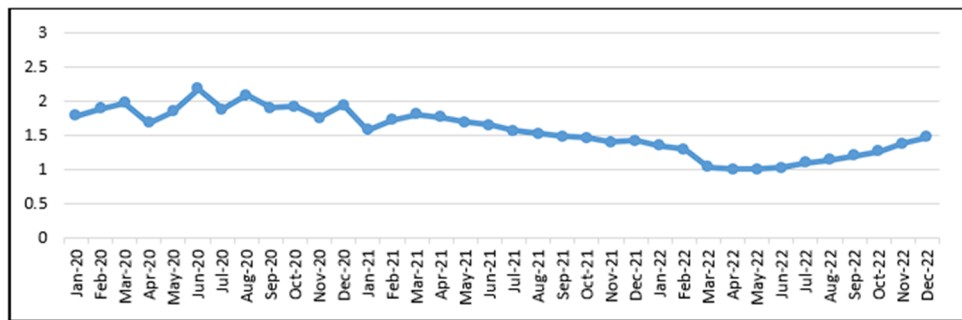

**Figure 15.** Monetary and financial sphere integral index dynamics.

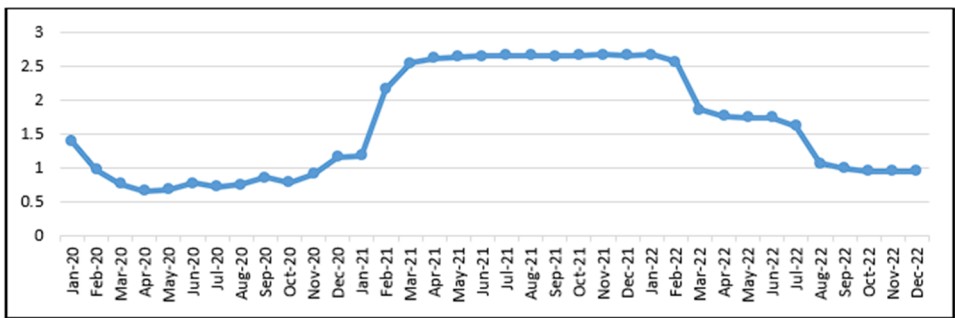

**Figure 16.** Foreign economic sphere integral index dynamics.

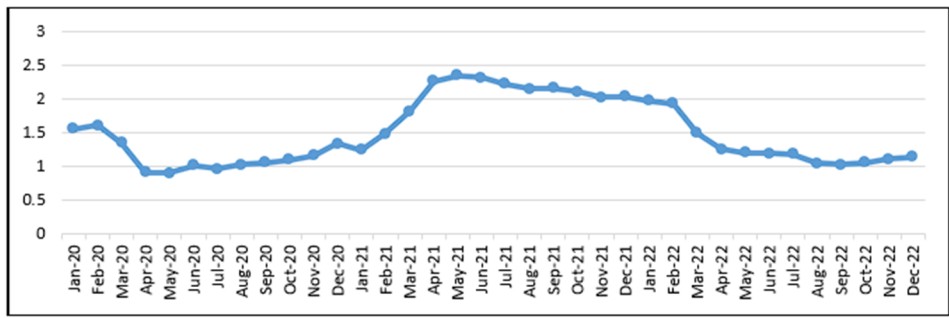

**Figure 17.** Generalized economic security index (Senchagov index) dynamics.

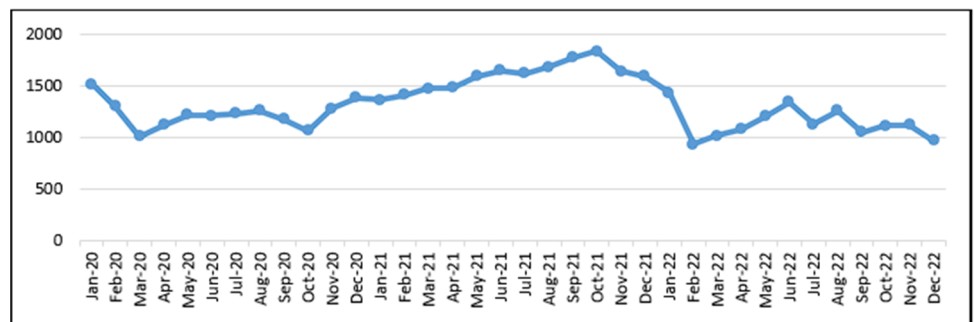

**Figure 18.** RTS index dynamics.

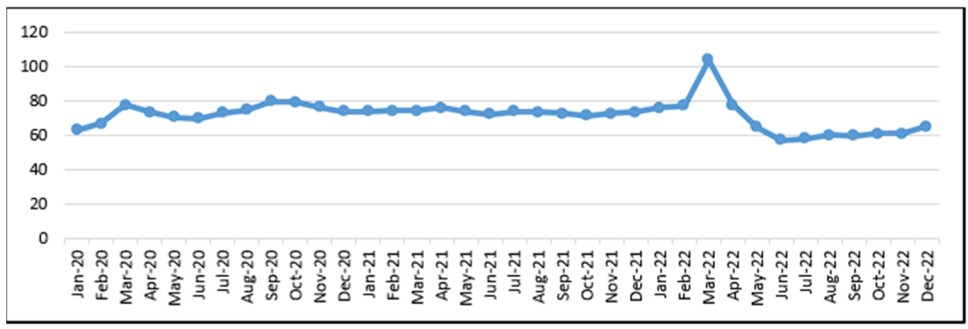

**Figure 19.** Ruble against dollar dynamics.

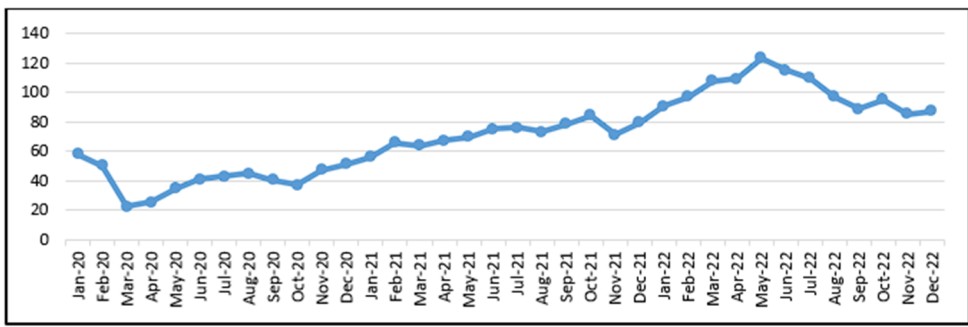

**Figure 20.** Oil price dynamics (USD/barrel).

Figure 21 shows the generalized OSEM index (Senchagov index) dynamics, obtained by us in this study, as well as in earlier works.

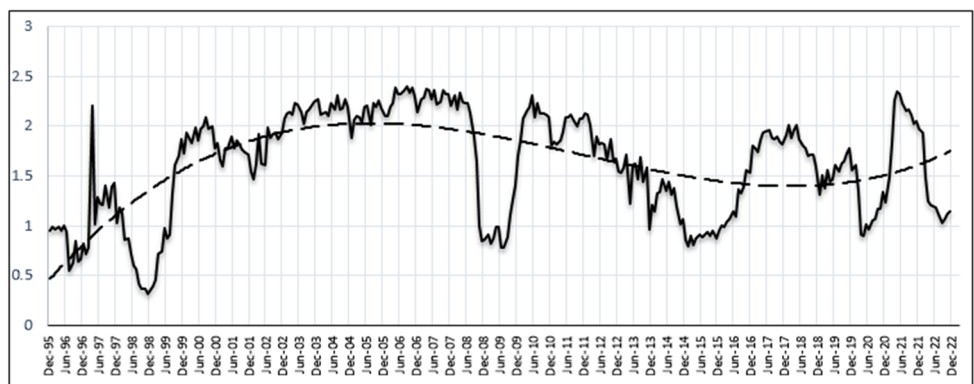

**Figure 21.** Generalized economic security index (Senchagov index) dynamics for the period from January 1996 to December 2022.

Analyzing the figure, the presence of a non-linear trend (dashed line) is noted, which indicates the economic security rapid indicators' unstable dynamics. The dashed line constructed by the least squares method using third degree polynomial. Data approximations are used to demonstrate general trends in the generalized economic security index, and can also be used to predict future values or analyze data changes over time. Negative impulses are clearly visible, which correspond to five economic crises. There were steady growth periods, when the index exceeded the value of 2, as well as economic crises periods, when the generalized index was less than 1. There is some concern about an increase in the frequency of economic crises.

Analysis of economic crises in Russia for the period 1996–2022 showed that all these crises had different causes and developed in different ways. The development of the last two crises is described in Figures 1–20. Let us consider the development features of earlier crises, clearly visible in Figure 21.

The crisis of 1998–1999 was accompanied by a significant decline in the economy's real sector indicators (GNP physical volume and industrial production indices) against the backdrop of extremely low levels of investment in fixed assets. In the social sphere, the decline in household income was accompanied by significant tension in the labor market, and a subsequent decline in consumer demand and retail trade turnover. In the monetary and financial sphere, with the turbulent nature of capital outflow, alongside gold and foreign exchange reserves imports' coverage being at a critical values level, a significant increase in prices was observed. The foreign economic sphere reacted with a significant decrease in exports and imports, as well as an increase in external debt, whose value was recovering within three years (Mityakov and Mityakov 2021).

The crisis of 2008–2009, which had different prerequisites and a global character, significantly affected the Russian economy. The crisis arose in the very center of the global financial system and was associated with problems in the United States' mortgage lending field. In Russia, the crisis was exacerbated by a large external debt presence and a sharp drop in oil prices. A significant amount of gold and foreign exchange reserves made it possible to avoid significant problems in the financial sector. The increase in the inflation rate was much smaller than in 1998–1999. At the same time, during this period, capital export from Russia increased. In the economy real sector, during the period 2008–2009, there was a deeper and longer drop in indicators than in 1998. The foreign economic sector was characterized by a deep decline caused by a reduction in exports and imports (Senchagov and Mityakov 2016).

The main prerequisites of the 2014–2016 crisis can be attributed to the following: stagnation in the economy, management dysfunctions, significant criminalization levels, excessive liberalization of the country's economy, and its raw material nature. The situation was aggravated by the introduction of economic sanctions in March–June 2015 and the fall in energy prices that began in June 2014. During the crisis, the decrease in the integral

indices' economic security values was recorded: to a lesser extent in the real economy, social, monetary and financial spheres, and to a greater extent in the foreign economic sphere. This crisis key feature is not in individual indicator's fall depth, but in a persistent long recession turning into stagnation. Mostly, economic parameters were restored only by the end of 2016.

In the process of each crisis analysis, factor chains are built, the interaction of which occur in certain circumstances, different for all crises. These factors are due to both external and internal factors. These include the immediate cause and the crisis cause (for example, the Government short-term bonds crisis in Russia or the mortgage crisis in the United States), as well as external and internal impacts on the system, including changes in exchange rates, energy prices, sanctions imposition and a number of others. As the crisis develops, the "domino model" is activated. A fall in one indicator is followed by a fall in a second, a third, and so on. The recovery phase may proceed in a different sequence. At the same time, the delay in the start of the corresponding parameter collapse, such as collapse duration and depth, as well as corresponding parameters of indicator restoration, can take completely different values in different cases.

At the same time, correlation and regression analysis methods, supplemented by elasticity theory, used to analyze the impact of Russia's economic security rapid indicators' economic crises, made it possible to formulate hypotheses about relationships between indicators. The first hypothesis was to change the statistical properties of economic security indicators during development and recovery periods from the crisis. The analysis showed that, during economic crises periods, processes described by the multiple regression model were largely deterministic (the average coefficients of determination were 0.841 in the period January 1997–August 1999 and 0.828 in the period August 2007–March 2010). During periods between crises, the processes described by the multiple regression model were largely stochastic, with determination coefficients from 0.3 to 0.6 (Senchagov and Mityakov 2013).

Another hypothesis is the presence of an external indicator—a crisis harbinger. From the data processing results for all five of the above crises, it was shown that there is such an indicator, and it is the RTS index, which acts as a harbinger, determining the beginning of a crisis. At the same time, the time delay between the start of RTS index growth (falling) and other economic security parameters ranges from one to several months. In our study, this earlier result is confirmed by 2020 and 2022 crises data. This is clearly seen from joint analysis in Figures 17 and 18. It is especially noticeable for the last crisis, which began in March 2022, where the RTS index fall was 5 months before. At the same time, the above facts do not provide grounds for the final confirmation of stock indices as economic crises harbingers' possibility usage hypothesis. Additional mathematical processing is required, which was not included in this study's goals and, most importantly, additional data are required that will allow the final solution to be found.

The proposed toolkit is unique and makes it possible to study socio-economic system parameters with the crisis development sampling frequency exceeding the characteristic periods (several months). This allows for retrospective and operational analysis of the economic security level and opens up opportunities for its forecasting. The technique is not absolutely new, since it was largely described in the authors' team head S.N. Mityakov's earlier works, which contain earlier crises analysis (Senchagov and Mityakov 2013, 2016; Mityakov and Mityakov 2021). At the same time, this article's scientific novelty is the use of this technique for analysis of the last two crises, as well as generalized economic security index dynamic analysis for the entire observation period.

Another debatable issue is index method feasibility. The use of individual indicators allows more detailed assessment of economic security dynamics in relation to the emergence of new threats. At the same time, the economic security individual spheres' integral indices and generalized index make it possible to monitor general trends in the system's development. In the case of applying the described methodology to countries and regions, the index method allows for a comparative analysis and finding their ratings by economic security levels.

### 5. Conclusions

The article proposes 12 main "rapid" indicators that can be used for the country's economic security operational monitoring, grouped into four spheres. The choice of indicators was dictated by sufficient coverage of certain economic security spheres. In addition, the proposed indicators were used based on availability of information from official sources, as well as information receipt required frequency. The non-linear transformations that were used made it possible to convert these indicators into a dimensionless form, which, in turn, made it possible to analyze indicators together and in dynamics. The use of integral indices made it possible to assess changes in the situation, aggregated according to economic security spheres and the country's economic security as a whole. The use of additional (exogenous) parameters made it possible to analyze them in comparison with indicators of economic security and study their interactions. In particular, one of the crises harbingers, apparently, is the RTS index. The proposed methodology is hardly universal and cannot be recommended for crisis analysis and forecasting in certain spheres, for example, education, healthcare, energy, etc. At the same time, general approaches are preserved here, so it is only necessary to change the indicators system.

During the study, the following results were obtained, which have scientific novelty:

1.  Using the proposed operational monitoring methodology and the index method, we built an economic security generalized index for the 1997–2022 period. This made it possible to identify the characteristic features of each crisis, as well as confirm the RTS index's usability as its harbinger.
2.  Detailed analysis of the dynamics of the last two economic crises was carried out using both individual and integral indices and Russia's economic security generalized index.

The proposed methodology and tools for economic systems' operational monitoring can be used effectively in government bodies' practical activities in order to ensure a proper level of economic security. The data available in official statistics, if interpreted effectively, can serve as a good tool for making managerial decisions. This is especially true for rapid diagnosis of crisis phenomena in countries and individual regions. At the country level, the study results can be claimed by various government agencies in the design of social and economic development strategic documents. At the regional level, these results will allow regions' leaderships to assess situations correctly and make scientifically based decisions on managing territories' economic growth in limited-resource conditions. The research subject's further development may be associated with additionally developing a toolkit, bringing machine learning and artificial intelligence data analysis methods. This, in our opinion, will make it possible to move to a qualitatively new level in understanding the development of the economy's crisis phenomena.

**Author Contributions:** Conceptualization, S.N.M.; Literature review, E.A.N.; Methodology, S.N.M.; Indicators system E.S.M.; Data processing, S.N.M.; Writing—initial draft preparation, A.I.L.; Writing—reviewing and editing, A.I.L.; Project administration, E.S.M. All authors have read and agreed to the published version of the manuscript.

**Funding:** This research was funded by the Russian Science Foundation (project No. 23-78-10009).

**Informed Consent Statement:** Not applicable.

**Data Availability Statement:** The data used is open-source. Neither classified nor trade secret data used. The data is available at http://www.gks.ru (accessed on 20 April 2023).

**Conflicts of Interest:** The authors declare no conflict of interest.

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
