# Peer review of "Country Economic Security Monitoring Rapid Indicators System"

_economies, doi:10.3390/economies11080208_

Round 1
Reviewer 1 Report
1) It is not known really what econometric or statistical analysis is done - and this is in my opinion enough to reject the paper and stop further peer review.
2) The paper is long, several sentences are long, complicated and hard to read.
3) Despite, it is really not clear what the paper is really about, which method is used etc.
4) Much specific language and terms are used, not easily approach by people after average economics and financial master studies. The paper should explain briefly such terms as this is paper for general audience, not a reasearch report for a narrow specialized group working in some topic
5) Equations are poorely described
6) Some parts contain only bullet points making it hard to understand what is going on in the paper
7) This is overall weak paper, long but lacking some hard analysis. It needs to be greatly revised before peer-review
Long complicated sententeces. Paper should be read by native English speaker.
Author Response
Dear reviewer!
Thank you for your comments, we’ve improved and rebuilt the article, following your concerns.
If you don’t mind, let us focus on main questions in detail.
- The article presents monitoring results, utilizing statistics from Russian officials, yet it was aggregated using presented formulas.
- We’ve dried the language and rewrote several sentences.
- We’ve highlighted the method used in body text: index method, on top of which we present generalized index methodology.
- We’ve tried to avoid specific terms and simplified the language overall in this revision.
- We’ve corrected descriptions for the equations.
- This particular section was corrected according to your remark.
- We’ve enhanced discussion section and added comprehensive analysis sector in the end of the article.
Once again, thank you for your comments and concerns regarding the article.
Reviewer 2 Report
This is a very interesting research article. It aims to develop a system of rapid – monthly – composite indicators of changes in regional or national economic security. Their index encompasses variables that are in the areas of real economy, social, financial, and international economy. They expand existing indicators into the period of the recent two crises, the Covid pandemic and the recent war and sanctions. The authors argue their system can be used for informed policy making. The paper is written carefully, and it cites a part of the relevant literature. The topic is important, and discussion on this topic should therefore be promoted.
That said, there are several points where the paper could be improved:
- The literature that the paper cites is primarily from authors from the former Soviet Union. Works from the rest of the world cited are not quantitative works. One should also consider other literature available or state there is none, which would be hard to argue. I would like to bring the attention of the authors to the work of Claudio Borio from the Bank for International Settlements on the prediction of financial crises, for instance.
- The incremental contribution compared to the previous works of the authors should be more clearly emphasized: which indicator, if any, is newly developed here, and which indicator is already developed earlier but is here calculated using new data form the two recent crises? What new finding is here compared to what we know from (their and others’) previous work. This should be clearly stated in several parts of the text. The authors should clearly state it even if the incremental contribution of the work is not large.
- I understand from their paper that there are some Russian national documents that seek to regulate the direction of focus on issues of economic security. Nevertheless, can we have an explanation of why those 12, and not some other indicators, are more relevant than others. Also, in which way does each of the indicators affect economic security – what is the exact mechanism? Importantly, to what extent does an absolute value matter (to avoid bankruptcy) and to what extent relative changes matter (to ensure people are not getting much less than they are accustomed to). Clearly, threats to national coming from economic security are different when observing different indicators.
- Target values and critical values should be justified. Are these taken from some other documents of governmental regulation, or economic literature? Alternatively, if one wants there thresholds to actually predict crises, one may want to determine them empirically, like in Borio and Lowe (2002).
- I am not persuaded what advantage does any aggregated indicator have over considering individual indicators, even if it means considering several of them. Individual indicators are clearer to interpret, and there is no arbitrariness in the weights of the individual indicators used. This is especially relevant because neither weights nor thresholds were chosen to optimize the prediction of crises (in the sense of a desirable noise-to-signal ratio or its predictive power) but based on some other, not clearly explained, considerations of the authors.
- The authors plot several indices. If I understand correctly, neither of these is developed in this study (unless the statement “[i]n this article, we adapted the index method for the ESOM problem” on page 8 means something else). Instead, they are just fed with most recent data that contains two major crises. They conclude by visual inspection that the RTS index can be used to conclude about the beginning and end of crises, and that it announces crises a few months earlier, and that this is confirmed by the recent crises. This raises several questions: can this be validated on other crises (in other countries for instance)? What types of crises can it predict? Is it difficult to imagine it can predict as well a foreign trade crises as well as a health crisis? Can we instead develop an indicator specifically for predicting crises, rather that developing aggregate indicators and then trying to conclude by inspection whether they accidentally also predict crises? Can we predefine and mark crises before we check whether an indicator predicts them? Is this composite indicator better at prediction than (some combination of the 12) individual indicators?
- What is the prospect i: a region of the country?
Language could be improved by using more active over passive, and by having fewer phrases constructed in a way which appears as directly translated from Russian but sounds unclear or unusual in English. For example: “to ensure economic security proper level” would sound better as “to ensure a proper level of economic security”.
On page 6 there is a paragraph that is mistakenly copied from the template or the instructions for the authors: “research manuscripts reporting large datasets …”.
Author Response
Dear reviewer!
We kindly thank you for your expanded and comprehensive review, please find our responses below.
- Firstly, we’ve reconsidered the literature and added several more western authors in order to maintain balance between Soviet authors and those from the West, including recommended Claudio Borio work.
- We’ve stated scientific novelty in the beginning and in the end of the article and highlighted which indices are new. Specifically, we state the novelty in conclusions (let me cite):
‘During the study, the following results were obtained, which have scientific novelty:
- Using proposed operational monitoring methodology and the index method, we built economic security generalized index for 1997-2022 period. This made it possible to identify each crises characteristic features, as well as to confirm RTS index usability as its harbinger.
- Last two economic crises dynamics detailed analysis was carried out both individual indicators and integral indices individual spheres and Russia's economic security generalized index.’
- These twelve indicators are included in economic security strategic documents, and have monthly discretization period, unlike the rest twenty eight (there are forty of them in strategies, accepted by Russian Government). Nevertheless, this dozen allow to adequately describe the whole economic system. We note, that significant importance have not only static, but dynamic indicators, which is easily seen from the statistics stated in the article.
- Critical and target values were taken from relevant scientific papers and Russian Economic Development Ministry indicators passport.
- We’ve highlighted in the reviewed article aggregated indicator power (in results and conclusion sections) and defined indices usage area. Specifically, it allows to carry out rapid analysis in order to accelerate decision-making.
- We’ve highlighted in the article once again that RTS index usage as crisis harbinger is hypothetical and needs to be confirmed within formal mathematical models and numerical experiments. We’ve added in the article, that in order to analyze specific crises we need the data and adopted indicators system, yet the proposed methodology is quite suitable.
- We’ve replaced this term with the term ‘sphere’, in order to avoid misunderstanding.
Once again, thank you very much for your patience and commentaries.
Reviewer 3 Report
The authors propose a rapid indicators system for monitoring economic activity. The system is based on monthly indicators and includes threshold values for analyzing various and forecasting crises for hierarchical levels of the economy.
The research motivations are solid, especially considering the COVID-19 pandemic. The proposal to monitor economic activity through high-frequency indicators is very interesting and useful for decision-makers.
I understand that the research needs improvement before its publication. My major concern is related to the theoretical framework insofar as it is decisive in evaluating the contributions and originality of the paper. Including nowcasting, literature is essential to enhance the contributions and highlight the paper's originality. See some references below:
• Big data types for macroeconomic nowcasting
• Google Econometrics: Nowcasting Euro Area Car Sales and Big Data Quality Requirements
• Macroeconomic nowcasting and forecasting with big data
• Massive data analytics for macroeconomic nowcasting
• Nowcasting and the Use of Big Data in Short Term Macroeconomic Forecasting: A Critical Review
• Nowcasting GDP and inflation: the real-time informational content of macroeconomic data releases
• Nowcasting macroeconomic indicators with alternative data
• Nowcasting unemployment insurance claims in the time of COVID-19
• Nowcasting unemployment rates with Google searches: evidence from the visegrad group countries
• Political preferences nowcasting with factor analysis and internet data: the 2012 and 2016 US presidential elections
Besides, the literature is heavily concentrated on the works of Senchagov, Mityakov E.S., and Mityakov S.N. This strong concentration weakens the paper and raises questions. How relevant is a topic that few authors discuss? Was the literature review carried out adequately? And etc.
Minor concerns
Present the concepts/definitions of economic terms used in the paper to avoid confusion, such as forecast versus prediction, index versus indicator, etc.
The reading of the paper would be more interesting and easier from the moment the authors adopt direct and concise language. In addition, the text lacks homogeneity. Terms such as sphere and perspective are used in the text as synonyms, making it difficult for the reader to understand the research.
Check the text between lines 293 and 297. Include the year of citations in lines 108, 176, 191, 196, and other similar cases. Check once more the direction between formulas 1 and 2.
Sincerely
The anonymous reviewer.
The paper lacks an extensive proofreading.
Author Response
Dear reviewer!
We kindly thank you for your review, please find our responses below.
Theoretical framework is vast and presented in cites from the literature and also partly included in this paper itself. We’ve tried to roll up most important steps and mathematical equations in the methodology section, yet, possibly it lacks explanations, but this is due the volume issues (the paper is long more than enough already).
We’ve highlighted scientific novelty of the article, especially in terms of indicators systems adoption and aggregated index methodology implementation at paper’s beginning and end. We’ve carefully read the article, simplified language and tried to avoid confusions in definitions.
Once again, thank you very much!
Round 2
Reviewer 1 Report
The description of the index constructions, its components weights, should be improved. Anyways, the paper was greately improved.
Author Response
Dear reviewer!
Thank you for the comments and remarks!
We've reconsidered index constructions' and its weights descriptions and added detailed explanation (highlighted with yellow).
Thank you very much!
Kind regards,
Author's collective
Reviewer 3 Report
Thanks. This version presents important improvements in relation to the first version. The authors expanded the literature, making the contributions clearer and well grounded.
The english is fine.
Author Response
Dear reviewer!
Thank you for your comments!
We've reconsidered index constructions' formulas, which weren't perfectly described both from meaning and leanguage viewpoints (highlighted with yellow). Also, we made minor improvements here and there.
Thank you very much!
Kind regards,
Author's collective